# Evaluating the appropriate oral lipid tolerance test model for investigating plasma triglyceride elevation in mice

**Masaru Ochiai** [ORCID]*

Department of Animal Science, School of Veterinary Medicine, Kitasato University, Towada, Aomori, Japan

* mochiai@vmas.kitasato-u.ac.jp

**Data Availability Statement:** All relevant data are within the manuscript files.

**Funding:** The author received no specific funding for this work.

## Abstract

The oral lipid tolerance test (OLTT) has been known to assess intestinal fat metabolism and whole-body lipid metabolism, but rodent models for OLTT are not yet established. Differences in OLTT methodology preclude the generation of definitive results, which may cause some confusion about the anti-hypertriglyceridemia effects of the test materials. To standardize and generate more appropriate methodology for the OLTT, we examined the effects of mice strain, dietary lipid sources, fasting period, and gender on lipid-induced hypertriglyceridemia in mice. First, lipid-induced hypertriglyceridemia was more strongly observed in male ddY mice than in C57BL/6N or ICR mice. Second, the administration of olive and soybean oils remarkably represented lipid-induced hypertriglyceridemia. Third, fasting period before the OLTT largely affected the plasma triglyceride elevation. Fasting for 12 h, but less than 48 h, provoked lipid-induced hypertriglyceridemia. Fourth, we explored the suppressive effects of epigallocatechin gallate (EGCG), a green tea polyphenol, on lipid-induced hypertriglyceridemia. The administration of 100 mg/kg of EGCG suppressed lipid-induced hypertriglyceridemia and intestinal lipase activity. Fifth, EGCG-induced suppressive effects were observed after lipid-induced hypertriglyceridemia was observed in male mice, but not in female mice. Lastly, lipid-induced hypertriglyceridemia could be more effectively induced in mice fed a high-fat diet for 1 week before the OLTT. These findings indicate that male ddY mice after 12 h fasting displayed marked lipid-induced hypertriglyceridemia in response to soybean oil. Hence, the defined experiment condition may be a more appropriate OLTT model for evaluating lipid-induced hypertriglyceridemia.

## Introduction

According to 2017 data from the World Health Organization, about 56 million people die per year worldwide [1]. Cardiovascular diseases have been identified as one of the most common causes of death globally. Postprandial hyperlipidemia and postprandial hyperglycemia are independent risk factors for cardiovascular diseases according to epidemiological evidence [2–9]. Coronary heart disease, type 2 diabetes, insulin resistance, and obesity are all associated with elevated postprandial plasma triglyceride (TG) levels [10, 11]. Recently, it was also

**Competing interests:** The author has declared that no competing interests exist.

suggested that a short-term high-fat feeding regimen in mice exacerbated postprandial plasma TG levels without altering fasting plasma TG levels [12]. Therefore, the improvement of post-prandial hypertriglyceridemia is supposedly a more valuable approach in lowering the risk of cardiovascular diseases than the improvement of fasting TG levels.

Postprandial plasma TG levels are strongly correlated with fasting triglyceride levels. How-ever, a difference in fasting TG levels only partially accounts for the interindividual variation in the magnitude of postprandial hyperlipidemia. The postprandial plasma TG response can be affected by genetic background, diet, physical activity, age, gender, and health conditions [13, 14]. In general, researchers have focused on fasting plasma TG levels, but not on postpran-dial levels in both human and rodent studies because they want to exclude the proximate effects of food materials on lipid metabolism, since health check programs in humans are often carried out under fasting conditions.

To examine the factors affecting postprandial hypertriglyceridemia and their mechanisms, an appropriate mice model of hypertriglyceridemia in response to dietary lipids is required. Information about the intestinal digestion and absorption of dietary fats is presented in Fig 1. The oral lipid tolerance test (OLTT) can assess not only intestinal lipid metabolism but also whole-body lipid metabolism. The ability to quickly normalize hyperlipidemia following the administration of lipids provides integrated information about intestinal lipid absorption, lipid transport via lipoproteins, and tissue-specific lipid metabolism. Thus, the OLTT is an essential and useful method to examine lipid metabolism. Importantly, data from the OLTT can be used to access information on the pancreas lipase inhibitory activity and intestinal lipid absorption when investigating the suppressive effects of food materials on postprandial hyper-triglyceridemia. However, rodent models for postprandial hypertriglyceridemia have not yet been standardized. As shown in Table 1, many researchers have used various models to exam-ine lipid-induced hypertriglyceridemia; specifically, the mice model, lipid dosage, lipid sources (lipids or lipids emulsions), and the fasting period before the OLTT often vary among studies, making direct comparisons difficult (Table 1). Various experimental protocols cannot generate the determinate results, which can cause confusion about the pharmaceutical effects of food materials. Fatty acids and dietary lipid composition and fasting period can largely affect lipid metabolism [15–17]. For example, when the anti-hypertriglyceridemia effects of diacylglycerol acyltransferase-1 inhibitors were investigated in mice, lipid sources, dosage, and the fasting period before the OLTT were not uniform [18–20] (Table 1). Furthermore, postprandial hyperglycemia has been identified as another well-known risk factor for coronary diseases, but its evaluation in mice and human models has been generally standardized [21], unlike in post-prandial hyperlipidemia testing. In humans, the glucose tolerance test is performed with the individual drinking a 75 g glucose solution following overnight fasting; in mice, glucose toler-ance is assessed through the oral administration of 2 g/kg glucose following 6 h of fasting [21].

The objective of this study was to determine the optimal protocols for the OLTT. We looked at variables such as mice strains, lipid sources, fasting period, and gender in mice.

## Materials and methods

The animal experimentation protocol was approved by the President of Kitasato University through the judgment of the Institutional Animal Care and Use Committee of Kitasato Uni-versity (Approval No. 19–194).

### Effect of mice model on lipids-induced hypertriglyceridemia (Exp 1)

Mice strains selected for the present OLTT experiment have been commonly used in the previ-ous studies (Table 1). Male ICR, ddY, and C57BL/6N mice were purchased from Japan SLC

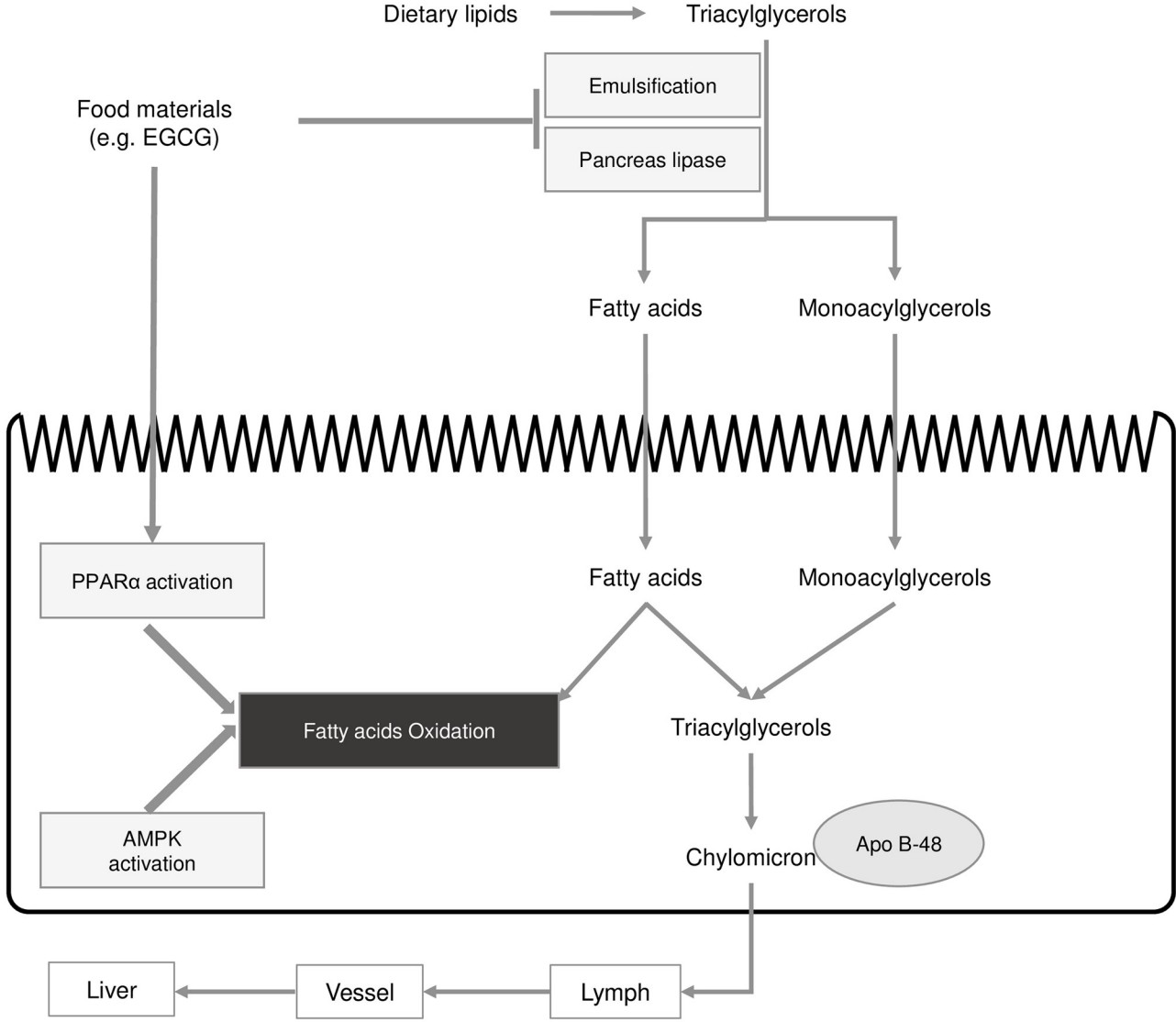

**Fig 1. Model of dietary lipid metabolism in small intestine.** Dietary lipids in digestive tract are emulsified and hydrolyzed by pancreas lipase to free fatty acids and monoglycerides. After the hydrolyzation, lipids are absorbed into small-intestinal epithelial cells, and followed by re-synthesized to TG. Chylomicrons are formed with synthesized TG and apolipoprotein B-48 and are transferred to blood via lymph. Some functional materials (e.g. EGCG) inhibit the emulsification and hydrolyzation of TG in diets, resulting to the suppression of lipids absorption. On the other hand, some functional materials activate the fatty acids β-oxidation through PPARα and AMPK activations.

(Hamamatsu, Japan) at 7 weeks of age. Mice (n = 10) were housed at 23 ± 2°C with lights on between 08:00 and 20:00. Food and water were accessed *ad libitum* (CE-2; Japan Clea, Tokyo) during a 2-week acclimation period. Mice were fasted for 12 h before the OLTT. Fasted mice were orally administered soybean oil [5 mL/kg; FUJIFILM Wako Pure Chemicals Corporation (Wako), Osaka, Japan]. Blood (30 μL/mice) was then collected from the tail vein and centrifuged (6,200×*g*, 4°C, 5 min) to obtain plasma prior to and at 60, 120, 180, 240, and 360 min after the administration of the oil. Plasma TG levels were immediately measured using a commercial kit (Wako). The area under the curve (AUC) values of plasma TG levels were calculated hourly using the trapezoidal rule.

**Table 1. List of OLTT models using mice or rats for evaluation of plasma TG levels.**

| Animal | | | | | Lipids used for OLTT | | | Fasting period | Sample | Ref. |
|---|---|---|---|---|---|---|---|---|---|---|
| Sp. | Strain | Diet | Gender | Age | Lipids | Emulsion | Dose | | | |
| M | C57BL/6J | HFD (2 w) | male | 10 w | Olive | - | 0.3 mL/head | 16 h | DHA | [31] |
| M | C57BL/6J | HFD (1 w) | male | 8–10 w | Coconut | - | 0.15 mL/head | 4 h | - | [12] |
| M | C57BL/6J | HFD (1 w) | male | 10 w | Olive | - | 0.3 mL/head | - | Bezafibrate | [46] |
| M | C57BL/6J | HFD (6 w) | male | 15 w | Olive | - | 10 mL/kg | 16 h | Inulin | [41] |
| M | C57BL/6J | chow | male | 6 w | Olive | - | 5 mL/kg | o.n. | Oolong tea and green tea | [47] |
| M | C57BL/6J | HFD (4 w) | male | 12 w | Olive | - | 17 mL/kg | o.n. | Pemafibrate | [48] |
| M | ICR | chow | male | 8 w | Olive and lard | - | 5 mL/kg | 20 h | Cocoa tea | [24] |
| M | ICR | - | - | 8 w | Corn | - | 5 mL/kg | 16 h | DGAT-1 inhibitor | [20] |
| M | ddY | - | male | 6 w | Olive | - | 10 mL/kg | - | Aged garlic | [49] |
| M | ICR | chow | male | | - | 20% soybean oil | 10 mL/kg | 14 h | AZD7687 (DGAT-1 inhibitor) | [18] |
| R | Wistar | chow | male | | Corn | - | 5 mL/kg | 14 h | | |
| M | C57BL/6J, C57BL/6N, ddY, etc. | chow | male | 8 w | Safflower | - | 0.4 mL/head | 24 h | - | [29] |
| M | C57BL/6, ICR etc. | | - | 5–9 w | Corn | - | 6 mL/kg | 16 h | A-922500 (DGAT-1 inhibitor) | [19] |
| R | SD, JCR | | - | 6–9 w | | - | | | | |
| R | SD | chow | male | 5 w | Corn | - | 5 mL/kg | o.n. | Resistant starch | [50] |
| R | Wistar | chow | male | 9 w | - | Soybean oil + lecithin + glycerol | 10 mL/kg | o.n. | Black tea polyphenol | [36] |
| R | Wistar | chow | male | 8 w | - | | | o.n. | Green tea catechin | [37] |
| R | Wistar | chow | male | 9 w | - | | | o.n. | Green tea catechin | [23] |
| R | SD | chow | - | 8 w | - | | | 18 h | Epsilon-polylysine | [38] |
| R | SD | chow | - | 9 w | - | High-oleic safflower oil + lecithin + glycerol | 10 mL/kg | o.n. | *Lactobacillus pentosus* | [39] |
| R | SD | chow | - | 5 w | - | 10% soybean oil | 15 mL/kg | o.n. | Tea leaves | [40] |

Sp., Species; M, Mice; R, Rat; -, not shown or unclear; o.n., overnight.

## Effect of lipids species on lipids-induced hypertriglyceridemia in mice (Exp 2)

Male ddY mice were obtained from Japan SLC at 6 weeks of age. Mice were acclimated for 2 weeks as described above. The ddY mice were fasted for 12 h before the OLTT. Mice were then divided into five groups (n = 8–10). Different oils were orally administered to each group of fasted mice. The dietary oils used in this study were olive oil, soybean oil, perilla oil, fish oil, and beef tallow (Wako). The fatty acid composition [C14:0, C16:0, C18:0, C18:1, C18:2, C18:3, C20:5 (EPA), and C22:6 (DHA)] of the test oils was determined using a gas chromatography (GC) system (GC2014, Shimadzu Co. Ltd., Kyoto, Japan) with a 60 m capillary column (TC70, GL Science Inc., Tokyo) and pure helium carrier gas, as described in the previous study [22]. The injector and detector temperatures of the GC equipment were 250 ˚C and 260 ˚C, respectively; the column oven temperature was constant at 180 ˚C for 40 minutes and was later increased by 20 ˚C/min from 180˚C to 260˚C. Before GC analysis, the fatty acids from the test oils were methyl-esterified using a commercial kit (Nacalai Tesque, Inc., Kyoto). The peak components were identified by comparing each retention time with that of a fatty acid methyl

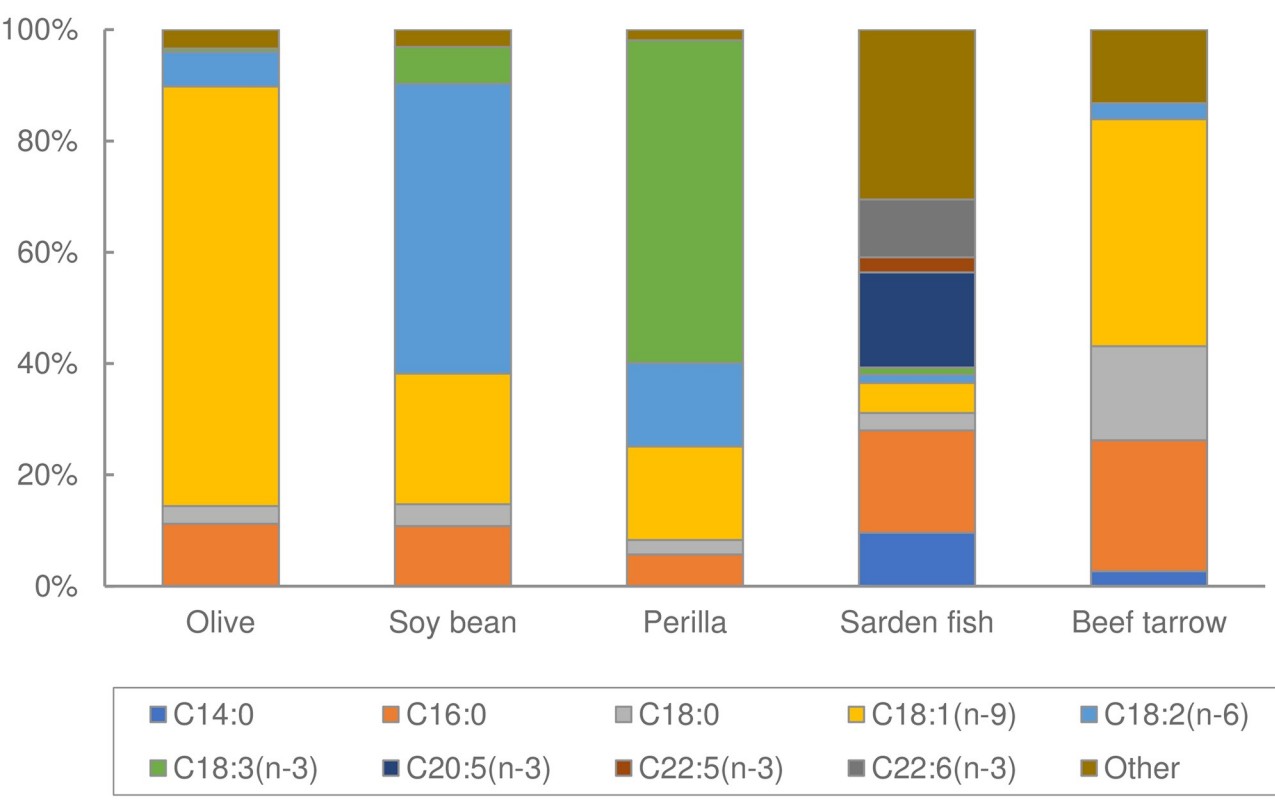

**Fig 2. Fatty acids composition of the test oils (Exp 2).**

ester (GLC-211, Funakoshi Co., Tokyo). The sum of the fatty acid percentages was estimated to be at 100%. The fatty acid composition of the oils is shown in Fig 2. Blood collection and the measurement of plasma TG levels and AUC values were carried out as described above (Exp 1). After completing Exp 2, mice were maintained on CE-2 for 4 weeks and were later used for Exp 3.

### Effect of fasting period on lipids-induced hypertriglyceridemia in mice (Exp 3)

Male ddY mice used in Exp 2 were used for another OLTT at 10 weeks of age. Mice were divided into five groups (n = 8–10). Before the OLTT, ddY mice were fasted for 0 (non-fasted), 3, 6, 12, and 48 h. Soybean oil (5 mL/kg) was orally administered. Blood collection and the measurement of plasma TG levels and AUC values were carried out as described above (Exp 1). After completing Exp 3, mice were maintained on CE-2 for 2 weeks and then used for Exp 4.

### Effect of green tea polyphenol, EGCG on lipids-induced hypertriglyceridemia in mice (Exp 4)

EGCG, a main polyphenol compound in green tea (*Camellia sinensis*) leaves, is an established food-derived pancreas lipase inhibitor known to suppress diet-induced hypertriglyceridemia in rodents [23, 24]. In Exp 4a, the suppressive effect of EGCG on lipid-induced hypertriglyceridemia was confirmed under the experimental conditions previously described in Exps 1–3.

The male ddY mice used in Exp 3 were used in Exp 4a at 12 weeks of age. Before the OLTT, the ddY mice were fasted for 12 h. Mice were divided into two groups (n = 8). Fasted mice were orally administered soybean oil (5 mL/kg) and water (5 mL/kg) (control group) or soybean oil (5 mL/kg) and 100 mg/kg of EGCG (purity ≧90.0%, Wako) (EGCG group). Blood collection and the measurement of plasma TG levels and AUC values were carried out as described above (Exp 1). After completing the OLTT, the mice were maintained on CE-2 for 1 week before they were used for biochemical analyses of lipid metabolism in the liver and small intestine (Exp 4b). Here, male ddY mice, age 13 weeks, were divided into two groups, with five mice per group. The five mice in each group were selected out of the eight mice on the basis of the lipid-induced elevation of plasma TG levels at 2 h during Exp 4a. Fasted mice were orally administered soybean oil (5 mL/kg) and water (5 mL/kg) (control group) or soybean oil and EGCG (100 mg/kg) (EGCG group). Two hours after administration, mice were euthanized under isoflurane anesthesia. The collected blood was centrifuged (6,200 g, 4°C, 15 min) to obtain plasma. The liver and small intestine were quickly removed and stored at −80°C until analyses were performed. The total lipids in the liver and the contents of the small intestine were extracted with a mixture of chloroform and methanol (2:1, v/v), according to the Folch method [25]. The liver TG content within the crude lipid extract was determined using a commercial kit (Wako). Extracted lipids in the small intestine were separated on a high-performance thin-layer chromatography plate (HPTLC; silica gel 60 plates, Merck, Germany) [26]. The plate was developed with a mixture of hexane/diethyl ether/acetic acid (60:40:1, v/v). The spots of each lipid (particularly TG, diacylglycerides, monoglycerides, and fatty acids) were visualized using iodine. As standards, reagent-grade triolein, diolein, monoolein, and oleic acid (Wako) were used. The activity of fatty acid synthase (FAS) in the liver was spectrophotometrically determined as described by Nepokroeff et al. [27]. For the extraction and crude purification of the enzymes, a small part of the liver was powdered in liquid nitrogen and homogenized in ice-cold Tris-HCL buffer containing 0.25 mol/L sucrose (pH 7.4). The total cytosol fraction was separated by centrifugation at $500 \times g$ for 10 min, followed by $9,000 \times g$ for 10 min and $12,000 \times$ g for 120 min. The enzymatic activity was measured at 30 °C and expressed as units per mg of wet tissue weight.

## Effect of gender on lipids-induced hypertriglyceridemia in mice (Exp 5)

Male and female ddY mice were obtained from Japan SLC at 6 weeks of age. Male and female ddY mice were similarly acclimated for a week as described above (Exp 1), and then, it was used for an OLTT at 7 weeks of age. The mice were divided by gender into groups: eight males and nine females. The mice were fasted for 12 h before the OLTT. Fasted mice were orally administered soybean oil (5 mL/kg) (control group) or soybean oil (5 mL/kg) with EGCG (100 mg/kg) (EGCG group). Blood collection and the measurement of plasma TG levels and AUC values were carried out as described above (Exp 1). After the completion of Exp. 5, the male mice were maintained on CE-2 for 1 week and then used for Exp 6.

## Effect of 1 week-feeding of high-fat and high-sucrose diet on lipids-induced hypertriglyceridemia in mice (Exp 6)

Male ddY mice used in Exp 5 were also used for Exp 6 at 8 weeks of age. The mice were fed with either low fat (LF) AIN-93G diet [7 wt% fat; 10 wt% sucrose] or the AIN-93G-based high-fat (HF) and high-sucrose diet [30 wt% fat; 20 wt% sucrose; F2HFHSD diet, Oriental Yeast Co., Ltd., Tokyo]. After feeding LF or HF diet, the mice in each diet group were divided into two groups (n = 8–9) for OLTT. The mice were then fasted for 12 h before the OLTT. Fasted mice were orally administered soybean oil (5 mL/kg) and water (5 mL/kg) (control group) or

soybean oil (5 mL/kg) and EGCG (100 mg/kg) (EGCG group). Blood collection and the measurement of plasma TG levels and AUC values were carried out as described above (Exp 1).

## Statistical analyses

All statistical analyses were performed using Excel Statistics 2015 (SSRI, Tokyo, Japan). A difference where $p < 0.05$ was considered statistically significant. In Exp 1, the data were expressed as mean ± SE (n = 8–10). The statistical analysis of differences among the three mice strains was performed using a one-way analysis of variance (ANOVA) and Tukey-Kramer test. In Exp 2, data are expressed as mean ± SE (n = 8–10). The statistical analysis of differences among the five dietary oil groups was performed using a one-way ANOVA and Tukey-Kramer test. In Exp 3, data are expressed as mean ± SE (n = 8–10). Statistical analysis of differences among the five fasting period groups was performed using a one-way ANOVA and Dunnett test, in which the fasting groups were compared to the non-fasting group. In Exp 4, the data were expressed as mean ± SE (n = 8). The statistical analysis of differences between the two groups was performed using the Student's t-test. In Exp 5, the data are expressed as mean ± SE (n = 8–9). Statistical analysis of differences among the groups was performed using a two-way ANOVA (gender and EGCG) and the Tukey-Kramer test. In Exp 6, the data are expressed as mean ± SE (n = 8–9). Statistical analysis of differences among the groups was performed using a two-way ANOVA (HF diet and EGCG) and the Tukey-Kramer test.

## Results

### An increase in lipids-induced hypertriglyceridemia in ddY mice (Exp 1)

The fasting plasma TG levels were found to be significantly higher in the ddY mice than in the other two strains (ICR, 78 ± 11 mg/dL; ddY, 142 ± 13 mg/dL; C57BL/6N, 60 ± 3 mg/dL). Plasma TG levels during the OLTT were also largely higher in ddY mice than in C57BL/6N and ICR mice. The AUC values of the plasma TG levels during the OLTT were also higher in the ddY mice than in C57BL/6N and ICR mice Fig 3.

### Olive and soybean oils leads to lipids-induced hypertriglyceridemia (Exp 2)

The AUC values of the plasma TG levels during the OLTT were found to be largely higher in ddY mice that were administered perilla oil, fish oil, and beef tallow than in ddY mice that were administered olive and soybean oils. The elevation of plasma TG levels at 60 and 120 min after oil administration was higher in mice administered olive and soybean oils Fig 4.

### A longer, but not too long, fasting leads to lipids-induced hypertriglyceridemia (Exp 3)

The elevation of plasma TG levels at 60 and 120 min after oil administration during the OLTT was significantly higher in ddY mice, particularly after 12 h fasting. Plasma TG levels at 180 min after oil administration during the OLTT were significantly higher in mice, which were fasted for 12 h or 48 h. The AUC values of the plasma TG level during the OLTT were higher in a fasting period-dependent manner Fig 5.

### EGCG suppresses lipids-induced hypertriglyceridemia in 12 h fasted ddY mice (Exp 4)

Plasma TG levels and AUC values during the OLTT were significantly lower in ddY mice that were administered 100 mg/kg EGCG Fig 6. The HPTLC data showed that spots of TG and

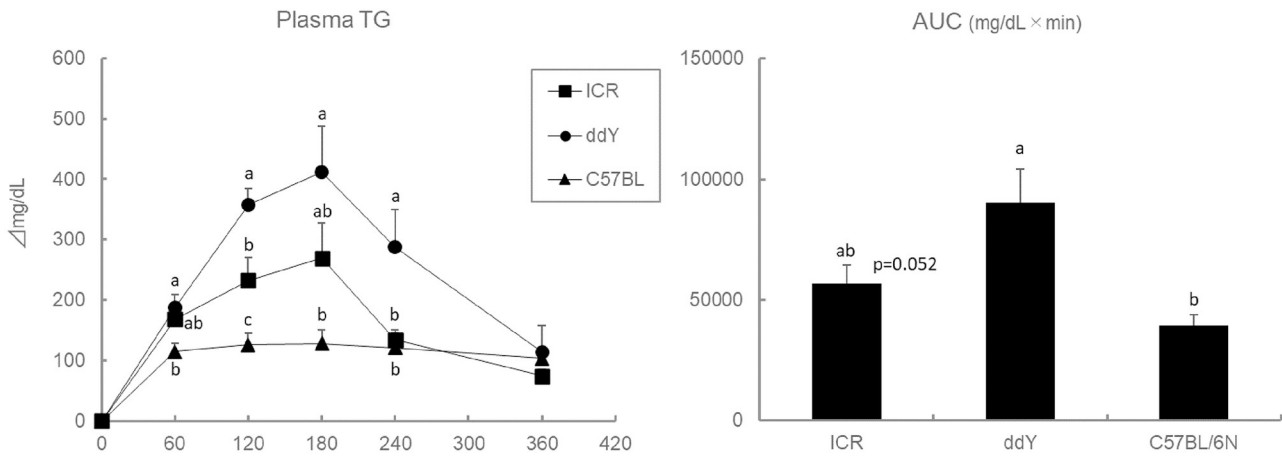

**Fig 3. Effect of mice strain on plasma TG levels and AUC values during the OLTT.** Values are means ± SE (n = 8–10). The data were analyzed with one-way ANOVA, followed by the post-hoc Tukey-Kramer test. Different letters are significantly different at $p < 0.05$.

diacylglycerides were clearly observed in the EGCG group, whereas those spots were not observed in the control group Fig 7. The plasma TG levels at 120 min after oil administration were significantly lower in the EGCG group, but the liver TG levels and FAS activity remained unchanged Fig 8.

## Lipids-induced hypertriglyceridemia is not fully induced in female ddY mice (Exp 5)

In male mice, the plasma TG levels and the AUC values observed during the OLTT were lower in the EGCG group (Fig 5). Conversely in female mice, the elevation of the plasma TG levels at 60 min after oil administration was lower in the EGCG group, but the maximal TG levels after oil administration and the corresponding AUC values were not changed by the presence of EGCG Fig 9.

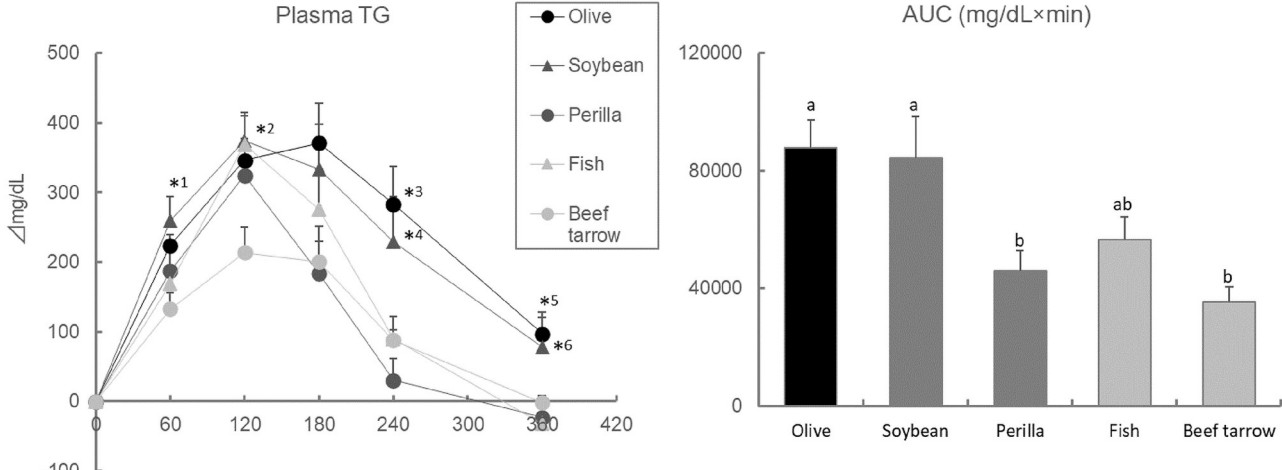

**Fig 4. Effect of lipid sources on plasma TG levels and AUC values during the OLTT in ddY mice.** Values are means ± SE (n = 8–10). The data were analyzed with one-way ANOVA, followed by the post-hoc Tukey-Kramer test. *: Asterisks are significantly different at $p < 0.05$ between the groups as below; *1, soybean—beef tarrow; *2, soybean, fish—beef tarrow; *3, perilla, fish, beef tarrow—olive; *4, perilla—soybean; *5, perilla, fish, beef tarrow—olive; *6, fish—soybean. Different letters in the AUC values are significantly different at $p < 0.05$.

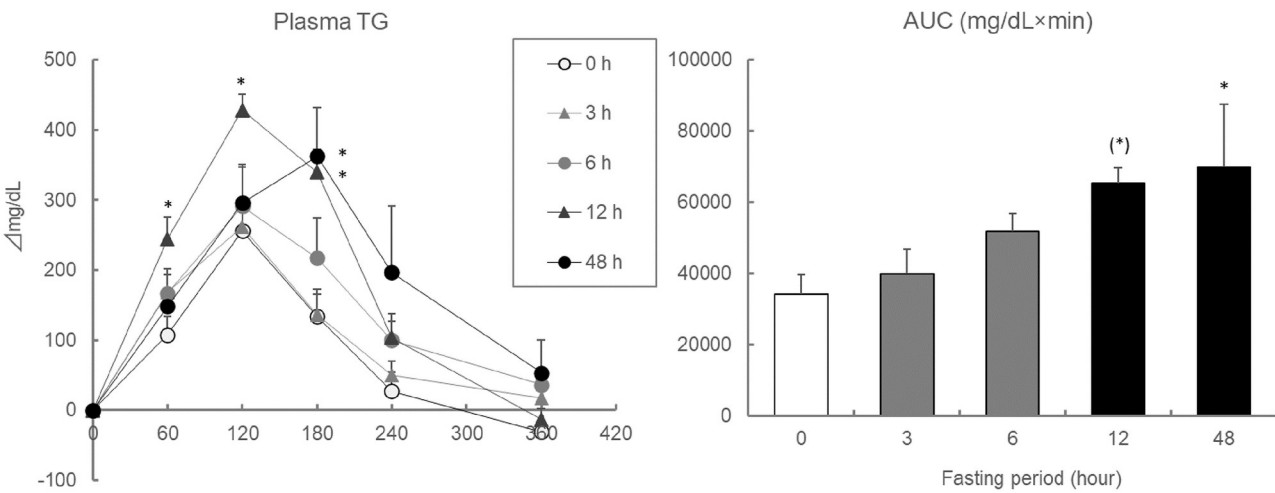

**Fig 5. Effect of fasting period on plasma TG levels and AUC values during the OLTT in ddY mice.** Values are means ± SE (n = 8–10). The data were analyzed with one-way ANOVA, followed by the post-hoc Dunnett test. *: asterisks show significant differences ($p < 0.05$) in compared to 0 h group. (*): (asterisks) show slight differences ($p < 0.1$) in compared to 0 h group.

## A short-term HF diet treatment induced lipids-induced hypertriglyceridemia (Exp 6)

The elevation of plasma TG levels after oil administration during the OLTT was significantly higher in ddY mice treated with HF diet for 1 week prior to testing than those fed with the AIN-93G control diet. Treatment with EGCG (100 mg/kg) suppressed lipid-induced plasma TG elevation and AUC values Fig 10.

## Discussion

The OLTT has been used to assess whole-body lipid homeostasis. The ability to quickly normalize hyperlipidemic metabolism following the oral administration of lipids provides integrated information about intestinal lipid digestion and absorption, lipoprotein transportation, and tissue-specific lipid metabolism. However, OLTT methodologies are highly varied and are

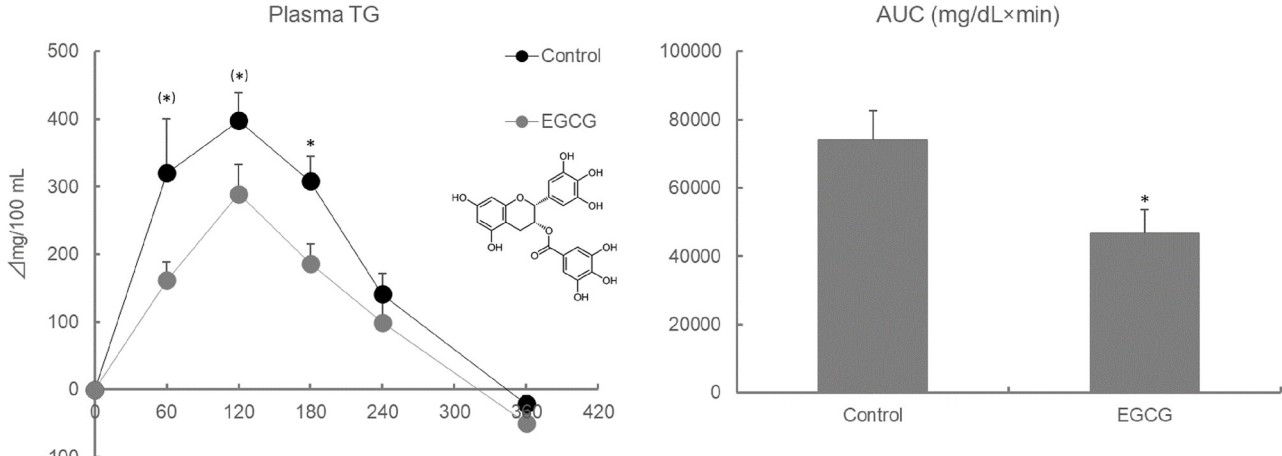

**Fig 6. Effect of EGCG on plasma TG levels and AUC values during the OLTT in ddY mice.** Values are means ± SE (n = 8). The data were analyzed with Student-t test. *: asterisks show significant differences ($p < 0.05$). (*): (asterisks) show slight differences ($p < 0.1$).

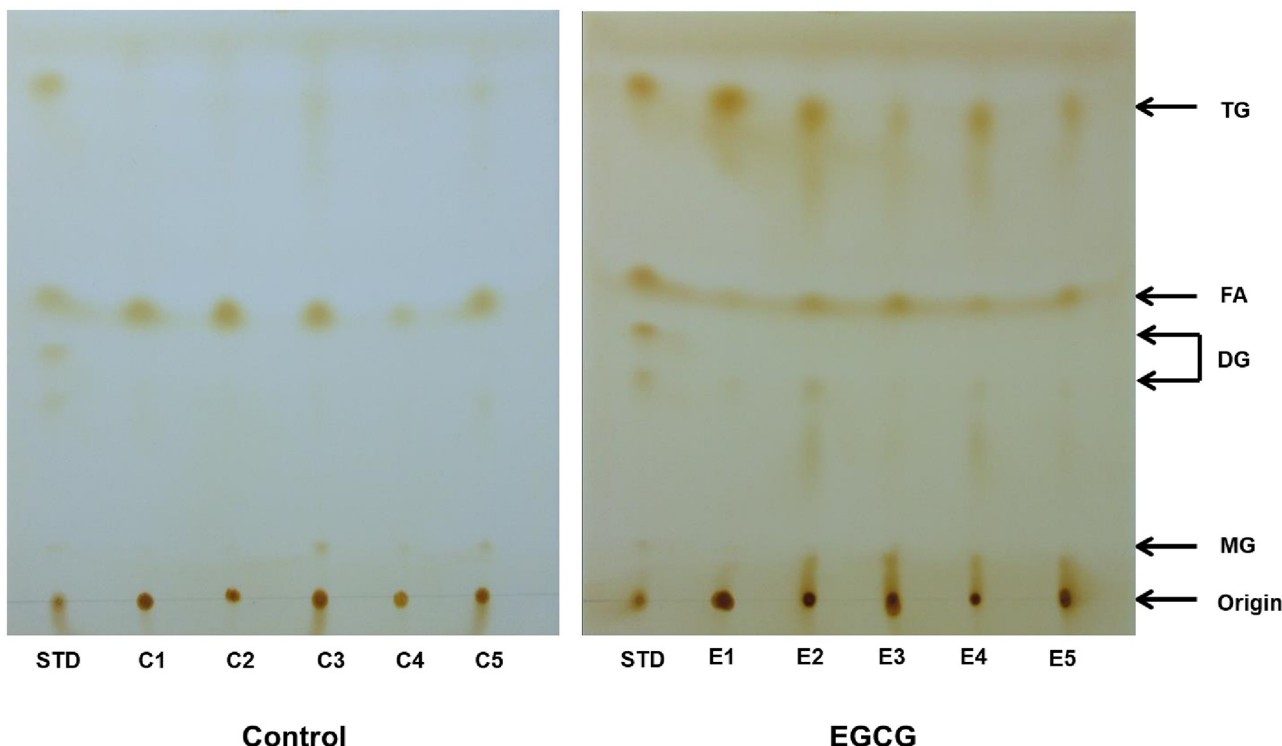

**Fig 7. Effect of EGCG on intestinal lipids source following the lipids administration in ddY mice.** STD contained a mixture of triolein, diolein, monoolein, and oleic acid. Extracted lipids in the small intestine were separated on a HPTLC. The plate was developed with hexane/diethyl ether/acetic acid (60:40:1, v/v). The spots of each lipid were visualized using iodine.

even different within research groups. Therefore, an appropriate model for the OLTT has not yet been constructed, whereas the methodology of the OGTT has been generally established in both mice [21] and humans by the WHO (2013) [28]. The wide array of available lipid sources and the complexity of lipid metabolism compared to glucose metabolism constitute two

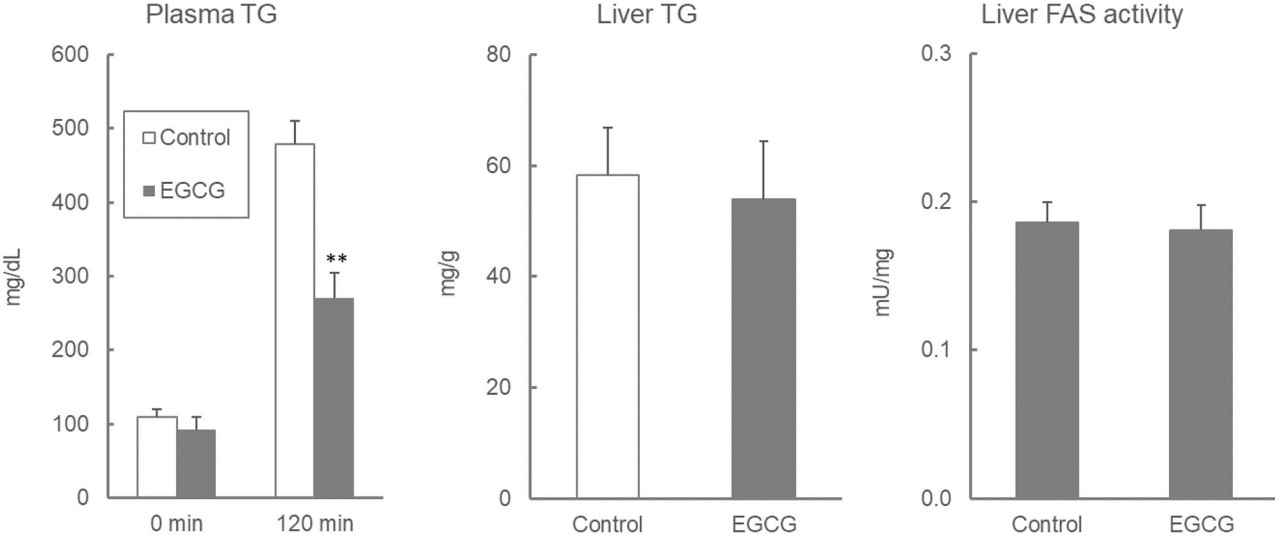

**Fig 8. Effect of EGCG on plasma and liver TG levels and liver FAS activity in ddY mice.** Values are means ± SE (n = 8). The data were analyzed with Student-t test. **: asterisks show significant differences ($p < 0.01$).

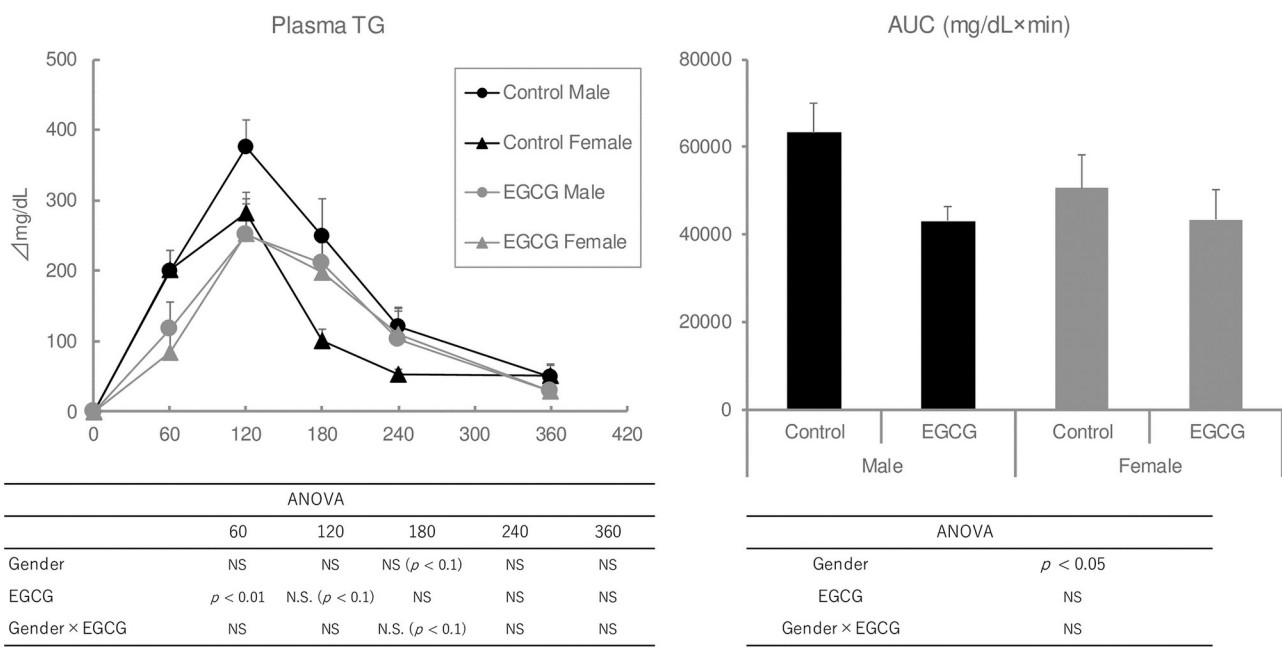

**Fig 9. Effect of gender on plasma TG levels during the OLTT in ddY mice.** Values are means ± SE (n = 8–9). The data were analyzed with two-way ANOVA, followed by the post-hoc Tukey-Kramer test. As statistically significances were not observed in two-way (gender and EGCG) ANOVA, a one-way ANOVA and comparison among the four groups was not carried out.

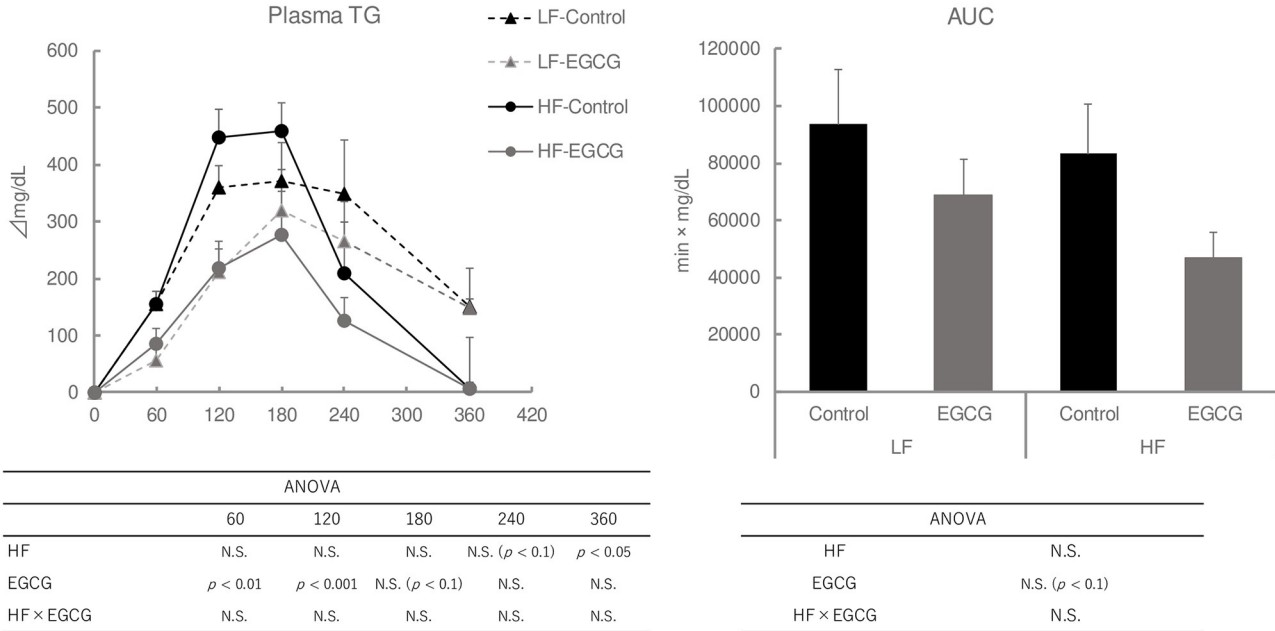

**Fig 10. Effect of short-term HFD on plasma TG levels during the OLTT in ddY mice.** Values are means ± SE (n = 8). The data were analyzed with two-way ANOVA, followed by the post-hoc Tukey-Kramer test. As statistically significances were not observed in the two-way (HF diet and EGCG) ANOVA, a one-way ANOVA and comparison among the four groups was not carried out.

reasons for the variation in OLTT methodologies. This present study has proposed a more appropriate model of the OLTT for evaluating lipid-induced hypertriglyceridemia in mice by investigating the differences between mice strains, lipid sources, the fasting period, gender, and diet before OLTT administration.

The first experiment of the present study confirmed that the plasma TG levels after the oral administration of lipids were different among the tested mice strains. Yamazaki et al. (2012) first reported that ddY mice were susceptible to lipid-induced hypertriglyceridemia due to an increase in lipoprotein production and a decrease in whole-body plasma lipoprotein lipase (LPL) activity [29]. The higher lipoprotein production and lower LPL activity indicate that chylomicron- and very-low-density-lipoprotein (VLDL)-TG in the plasma cannot be hydrolyzed and transported into the tissues, which often results in hypertriglyceridemia. Saleh et al. (2011) also suggested in their review that LPL may be a causative factor for the acceleration of lipid clearance from the blood [30]. However, ddY mice are known to possess higher LPL activities and have the potential to induce lipid-induced hypertriglyceridemia; however, they are not often selected for OLTT studies (Table 1). The increases in the plasma TG levels 180 min after oil administration in the C57BL/6N and ICR mice were 128 and 270 mg/dL, respectively, which were remarkably lower than in the ddY mice (412 mg/dL). Considering the smaller increase in plasma TG levels observed in the C57BL/6N and ICR mice, it is likely more difficult to evaluate the suppressive effects of food materials on hypertriglyceridemia in these strains. Correspondingly, in previous studies that used C57BL/6 mice strain, the administration of lipids elevated the plasma TG levels only to about twice the value of the fasting baseline. Yamazaki et al. (2012) reported that the C57BL/6 mice strain did not show postprandial hypertriglyceridemia [29]. Therefore, we contend that the choice of mice strain used for the OLTT is an important factor for the evaluation of lipid-induced hypertriglyceridemia.

The lipid source used for the OLTT is considered to be an important factor for lipid-induced hypertriglyceridemia. In rodent studies, olive and corn oils have often been used as the lipid source (Table 1). In human studies, various high-fat meals were utilized as the lipid source. The dominant fatty acids in olive and corn oils are oleic acid (rich in n-9 fatty acids) and linoleic acid (rich in n-6 fatty acids), respectively, but not n-3 fatty acids (Fig 1). In this study, dietary oils rich in n-6 and n-9 fatty acids could easily elevate the plasma TG levels. It has been reported that dietary n-3 fatty acids such as α-linolenic acid, eicosapentaenoic acid (EPA), and docosahexaenoic acid (DHA) improved lipid metabolism in rodents and humans through anti-inflammatory and peroxisome proliferator-activated receptor (PPAR)-α/γ activation mechanisms [31]. The activation of PPAR-α by n-3 fatty acids can increase fatty acid oxidation and decrease TG and VLDL secretion. Furthermore, the activation of PPAR-γ by n-3 fatty acids improves insulin sensitivity, resulting in the increase of TG clearance [15, 16]. The present results indicate that fish and perilla oils rich in n-3 fatty acids decreased the plasma TG levels and accelerated TG clearance at 180 min after oil administration, although the maximum plasma TG levels were not suppressed. These results indicate that n-3 fatty acids accelerated fatty acid oxidation. Additionally, n-3 fatty acids were identified to be less susceptible to pancreatic lipase activity due to the presence of multiple double bonds and their overall structural complexity, which may delay their digestion and absorption [32–35]. In contrast, the administration of beef tallow, which is rich in saturated fatty acids, unexpectedly suppressed plasma TG elevation. In general, saturated fats such as those found in beef tallow and lard can easily induce abnormalities in glucose and fat metabolism, delaying TG clearance. On the other hand, saturated fats are less susceptible to pancreatic lipase activity, in part due to their higher melting points, resulting in lower fat absorption. Therefore, these fats can take longer time to be digested and absorbed in the small intestine, and the plasma TG levels can be suppressed after the administration of saturated fats. Due to the higher melting points of saturated

fats, they are more difficult to orally administer to rodents with a stainless-steel tube at room temperature. On the other hand, saturated fats have often been used for the OLTT in human studies. This must be carefully considered when comparing the results between mice and human studies. This study aimed to propose an appropriate dietary lipid for OLTT that will increase the plasma TG elevation in mice, and we determined that soybean oil is the most suitable lipid source for this purpose. Considering the suppression of the plasma TG elevation and AUC values by n-3 fatty acids, oils rich in n-3 fatty acids are not likely to be appropriate for OLTT.

Dietary lipid size can affect the digestion and absorption of lipids in the small intestine, which can easily elevate the plasma TG levels. As shown in Table 1, fat emulsions have often been utilized as fat sources when considering the intestinal digestion and absorption of lipids [18, 23, 36–40]. Generally, dietary lipids are emulsified by endogenous bile acids to smaller lipid droplets in the gut, and then they are digested and absorbed in the small intestine. Administering the mixture of oils and emulsifiers has been observed to accelerate lipid digestion. Some reports have shown that the plasma TG levels reached their maximum 2–3 hours after the administrations of fat emulsions [23, 36–40], which is similar to the timeframe observed in this study and other reports in which no emulsions were used [24, 29, 41]. Higher maximum values of plasma TG levels during the OLTT are needed for easier evaluation of hypertriglyceridemia; hence, different methodologies for lipid preparation should be investigated.

The fasting period before the OLTT can greatly affect the resulting plasma TG elevation, but few investigations examining the effects of fasting period before the OLTT on plasma TG elevation have been carried out [17, 42]. Ikeda et al. (2014) demonstrated that a longer fasting period dramatically suppressed mRNA expression associated with lipogenesis but activated mRNA expression associated with lipid β-oxidation in the liver [17]. The fasting period-dependent increase in the AUC values of plasma TG levels suggests that the plasma TG levels did not reach the maximum and that orally administered lipids were effectively absorbed. In the case of 48 h-fasting period, the maximum plasma TG elevations were found to be delayed, indicating that the administered lipids were digested and absorbed in the small intestine but were quickly transported into the liver, before they could enter the blood and be detected in the plasma. Therefore, fasting treatment for 12 hours before the OLTT is more suitable for the evaluation of plasma TG elevation as a fasting period that is too long before an OLTT can greatly affect lipid metabolism; this suggests that a 12 h fasting period is more appropriate than a 48 h fasting period.

EGCG is known to have multi-faceted effects [43]. In Japan, EGCG is often utilized as a functional compound in health foods, supplements, and drinks. The inhibition of pancreatic lipase activity and the suppression of dietary lipid absorption are considered to be anti-obesity mechanisms of EGCG [23]. In this study, the administration of EGCG (100 mg/kg) strongly suppressed the plasma TG elevation during the OLTT, and HPTLC data of small intestinal contents showed that TG remained in the small intestine and had not been degraded. EGCG suppressed the elevation of plasma TG to 162, 289, and 186 mg/dL compared to the control (320, 398, and 308 mg/dL) at 60, 120, and 180 min after administration, respectively. Therefore, the investigated protocols (ddY, male, 12 h fasting before OLTT, and soybean oil) have been determined to be appropriate for the evaluation of food-derived compounds in OLTT. Considering the results from Fig 2 (Exp 1), it is not expected that the C57BL/6N and ICR mice represent a strong suppressive effect of EGCG on hypertriglyceridemia as observed in the ddY mice. On the other hand, a single administration of EGCG did not affect TG content and FAS activity in the liver, unlike in the plasma.

Gender has a considerable influence on lipid-induced hypertriglyceridemia. Many reports have used male mice or rats for evaluating the effects of food materials on lipid-induced hypertriglyceridemia in OLTT studies (Table 1), although the reasons why male rodents were exclusively used remain unclear. Some review articles have indicated that male mice are more susceptible to postprandial hypertriglyceridemia than female mice by some endogenous factors [30, 44]. Saleh et al. (2011) and Murray et al. (1999) pointed out that acylation-stimulating protein (ASP), which is produced by adipocytes, accelerated postprandial TG clearance in women and that a significant association between progesterone and ASP levels contributed to abdominal fat accumulation in women [30, 44]. In rodents, gender dimorphism was observed in the postprandial response; female mice displayed larger increases in adipose tissue weights and LPL activities compared to male mice. The rapid clearance of lipids in female mice was suggested to be caused by LPL activity. Our findings indicate that the maximum plasma TG levels during OLTT were higher in male mice and EGCG suppressed the levels in male mice only. Therefore, in the case of OLTT investigations using female mice, the suppressive effects of food materials on plasma TG elevation may be difficult to be evaluated.

Finally, the implementation of high-fat diets for over several weeks exacerbates metabolic syndrome parameters observed in the plasma and tissues of mice. These diets could not only exacerbate dysfunctional lipid metabolism, but also exacerbate glucose intolerance and insulin resistance. On the other hand, a short-term regimen of a high-fat diet (1 week) can induce lipid-induced hypertriglyceridemia without affecting fasting plasma TG levels [12, 31]. Hernández Vallejo et al. (2009) suggested that 1-week adaptation to a saturated fat-based high-fat diet could induce postprandial hypertriglyceridemia, but it could not affect the fasting plasma TG levels, by inducing intestinal TG synthesis and decreasing chylomicron secretions [12]. In particular, the serum apo-B48 levels in a fasted state can be a useful marker of postprandial hypertriglyceridemia [45], which may be another method to associate lipid-induced TG elevation in the OLTT.

## Conclusions

This study compared TG responses after the oral administration of various dietary lipids in several strains of fasting mice. These findings helped elucidate more appropriate OLTT models. We determined that male ddY mice fasted for 12 h displayed markedly higher lipid-induced hypertriglyceridemia in response to soybean oils rich in n-6 fatty acids. Lipid-induced hypertriglyceridemia and postprandial hyperglycemia are determined to be independent risk factors for coronary diseases. Determining standard protocols for lipid-induced hypertriglyceridemia testing is requisite for investigating lipid metabolism in mice.

## Supporting information

**S1 Fig. HPTLC chromatogram of the intestinal content in mice administered with vehicle or EGCG.** The S1 Fig supports Fig 7. The S1 Fig shows the effect of EGCG on intestinal lipids source following the lipids administration in ddY mice. Extracted lipids in the small intestine were separated on a HPTLC. STD, a mixture of standard lipids, contained a mixture of reagent-grade triolein (TG), diolein (DG), monoolein (MG), and oleic acid (FA). The spot was developed on a plate with hexane/diethyl ether/acetic acid (60:40:1, v/v). The developed spots of each lipid were visualized using iodine.
(PDF)

## Acknowledgments

I thank Ms. Serina Yamashita and Ms. Ao Matsuki (Kitasato University) for technical assistants in animals care. We thank Enago (https://www.enago.jp/) for English language editing.

## Author Contributions

**Conceptualization:** Masaru Ochiai.

**Data curation:** Masaru Ochiai.

**Formal analysis:** Masaru Ochiai.

**Investigation:** Masaru Ochiai.

**Validation:** Masaru Ochiai.

**Writing – original draft:** Masaru Ochiai.

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
