## [Decision Letter · Decision Letter 0]

25 Aug 2020

PONE-D-20-18913

Evaluating the appropriate oral lipid tolerance test model for investigating plasma triglyceride elevation in mice

PLOS ONE

Dear Dr. Ochiai,

Thank you for submitting your manuscript to PLOS ONE. After careful consideration, we feel that it has merit but does not fully meet PLOS ONE’s publication criteria as it currently stands. Therefore, we invite you to submit a revised version of the manuscript that addresses the points raised during the review process.

We look forward to receiving your revised manuscript.

Kind regards,

Ouliana Ziouzenkova, PhD

Academic Editor

PLOS ONE

Journal Requirements:

Reviewers' comments:

Reviewer's Responses to Questions

**Comments to the Author**

1. Is the manuscript technically sound, and do the data support the conclusions?

Reviewer #1: Yes

Reviewer #2: Yes

2. Has the statistical analysis been performed appropriately and rigorously? 

Reviewer #1: Yes

Reviewer #2: Yes

3. Have the authors made all data underlying the findings in their manuscript fully available?

Reviewer #1: Yes

Reviewer #2: Yes

4. Is the manuscript presented in an intelligible fashion and written in standard English?

Reviewer #1: Yes

Reviewer #2: Yes

5. Review Comments to the Author

Reviewer #1: Authors evaluated rodent models for OLTT and then suggested an optimal method.

Basically, the reported models would be fine because pharmacological substances would always be evaluated compared to controls. However, the larger the dynamic rage, the clearer the results from the bioassay. OTLL models were evaluated under a variety of conditions, and the suggested method could be helpful to researchers.

The publication of this manuscript in PLOS ONE can be supported after the suitable modification as follows.

-Abstract need to be concise and written shorter.

-Page 7, line 103, “I selected….” should be rewritten. Do not use “I”.

-All figures should be provided as high-resolution images.

-Mark statistical significance on the graph.

-In figure 2, please make it easy to distinguish.

Reviewer #2: The manuscript entitled “Evaluating the appropriate oral lipid tolerance test model for investigating plasma triglyceride elevation in mice” have described the complete standardization of a novel model to evaluate the postprandial hyperlipidemia trough oral lipid tolerance test, demonstrating which is the best strain, the optimum oil and the difference between gender...

The study was written carefully and well in terms of language. However, this work requires minor corrections.

Minor issues:

- Ethically it is an excellent idea to use the same animals for different experiments, however, in this case a method is being standardized and the age must always be the same to be exact. I suggest that in the future this be done with animals of the same age in order to confirm the results obtained in your model.

- In the high-performance thin-layer chromatography plate section in line 169-172 i suggest to mention which standard was used to compare and determine the lipids.

6. PLOS authors have the option to publish the peer review history of their article (what does this mean?). If published, this will include your full peer review and any attached files.

Reviewer #1: No

Reviewer #2: No

---

## [Author Response · Author response to Decision Letter 0]

17 Sep 2020

The author very appreciated to the precise comments, questions, and suggestions to the manuscript. The author has prepared the reply point by point as also attached in the submission file.

Reviewer #1: The publication of this manuscript in PLOS ONE can be supported after the suitable modification as follows.

-Abstract need to be concise and written shorter.

Reply: The previous manuscript observed a PLOS ONE guideline of the abstract section (not exceed 300 words), but the abstract has been more concisely revised.

-Page 7, line 103, “I selected….” should be rewritten. Do not use “I”.

Reply: The part pointed out has been revised (Line 104).

-All figures should be provided as high-resolution images.

Reply: The figures have been clearly revised.

-Mark statistical significance on the graph.

Reply: I think the reviewer pointed out the Fig 9 and Fig 10. The part pointed out has been explained in detail (Fig 9-10 captions). In case that interactions between the two factors, a comparison among the four groups would be performed using a one-way-ANOVA and Tukey-Kramer test. However, because statistically significances were not observed in the two-way ANOVA, a one-way ANOVA was not carried out according to statistical analysis rules. (Line 294, 306)

-In figure 2, please make it easy to distinguish.

Reply: The part pointed out has been clearly revised. Figure 2 has been changed to be a color mode.

 

Reviewer #2: ~ ~ ~ However, this work requires minor corrections.

- I suggest that in the future this be done with animals of the same age in order to confirm the results obtained in your model.

Reply: I appreciated to the comment. In this study, various aged mice models were used for the investigation because normal model mice were not in the special pathological condition. When the suppressive effect of food materials on lipids-induced hypertriglyceridemia is evaluated in the future, the same aged ddY mice should be used.

- In the high-performance thin-layer chromatography plate section in line 169-172. I suggest to mention which standard was used to compare and determine the lipids.

Reply: The part pointed out has been revised (Line 174).

---

## [Editor Report · Decision Letter 1]

23 Sep 2020

Evaluating the appropriate oral lipid tolerance test model for investigating plasma triglyceride elevation in mice

PONE-D-20-18913R1

Dear Dr. Ochiai,

We’re pleased to inform you that your manuscript has been judged scientifically suitable for publication and will be formally accepted for publication once it meets all outstanding technical requirements.

Kind regards,

Ouliana Ziouzenkova, PhD

Academic Editor

PLOS ONE

---

## [Editor Report · Acceptance letter]

25 Sep 2020

PONE-D-20-18913R1 

Evaluating the appropriate oral lipid tolerance test model for investigating
 plasma triglyceride elevation in mice 

Dear Dr. Ochiai:

I'm pleased to inform you that your manuscript has been deemed suitable for publication in PLOS ONE. Congratulations! Your manuscript is now with our production department. 

Kind regards, 

on behalf of

Dr. Ouliana Ziouzenkova 

Academic Editor

PLOS ONE